# Predicting Structural Consequences of Antibody Light Chain N-Glycosylation in AL Amyloidosis

**DOI:** 10.3390/ph17111542

**Published:** 2024-11-16

**Authors:** Gareth J. Morgan, Zach Yung, Brian H. Spencer, Vaishali Sanchorawala, Tatiana Prokaeva

**Affiliations:** 1Boston University Amyloidosis Center, Boston University Chobanian & Avedisian School of Medicine, Boston, MA 02118, USA; 2Section of Hematology and Medical Oncology, Boston University Chobanian & Avedisian School of Medicine, Boston, MA 02118, USA; 3Department of Pathology and Laboratory Medicine, Boston University Chobanian & Avedisian School of Medicine, Boston, MA 02118, USA

**Keywords:** light chain amyloidosis, immunoglobulin, systemic amyloid disease, N-glycosylation, sequon, amyloid fibrils, protein misfolding, protein aggregation

## Abstract

**Background/Objectives:** Antibody light chains form amyloid fibrils that lead to progressive tissue damage in amyloid light chain (AL) amyloidosis. The properties of each patient’s unique light chain appear to determine its propensity to form amyloid. One factor is N-glycosylation, which is more frequent in amyloid-associated light chains than in light chains from the normal immune repertoire. However, the mechanisms underlying this association are unknown. Here, we investigate the frequency and position within the light chain sequence of the N-glycosylation sequence motif, or sequon. **Methods:** Monoclonal light chains from AL amyloidosis and multiple myeloma were identified from the AL-Base repository. Polyclonal light chains were obtained from the Observed Antibody Space resource. We compared the fraction of light chains from each group harboring an N-glycosylation sequon, and the positions of these sequons within the sequences. **Results:** Sequons are enriched among AL-associated light chains derived from a subset of precursor germline genes. Sequons are observed at multiple positions, which differ between the two types of light chains, κ and λ, but are similar between light chains from AL amyloidosis and multiple myeloma. Positions of sequons map to residues with surface-exposed sidechains that are compatible with the folded structures of light chains. Within the known structures of λ AL amyloid fibrils, many residues where sequons are observed are buried, inconsistent with N-glycosylation. **Conclusions:** There is no clear structural rationale for why N-glycosylation of κ light chains is associated with AL amyloidosis. A better understanding of the roles of N-glycosylation in AL amyloidosis is required before it can be used as a marker for disease risk.

## 1. Introduction

Amyloid fibrils are insoluble protein aggregates with a characteristic appearance under the electron microscope [1]. These fibrils are best known as the hallmark of several human diseases, which are collectively known as amyloidosis [2,3]. Amyloid fibrils are most often homopolymers of a single precursor protein, held together in a cross-β structure by intermolecular hydrogen bonding between backbone amides. The precursor proteins for most human amyloid diseases are secreted from their cell of origin into circulation before aggregating in distal tissues. Amyloid precursors are often covalently modified after translation, and these post-translational modifications (PTMs; see the list of abbreviations) may be incorporated into the structure of the amyloid fibril [4,5]. Commonly observed modifications in secreted proteins include disulfide bonds, proteolytic cleavage and N-glycosylation [6]. The effects of such modifications on the propensity of precursor protein to form amyloid fibrils is not well understood. Here, we examine the role of N-glycosylation, which is the covalent attachment of sugar molecules (glycans) to the amide side chains of asparagine residues, in amyloid light chain (AL) amyloidosis.

AL amyloidosis is caused by the aggregation of antibody light chain (LC) proteins, or their fragments, as amyloid fibrils in multiple tissues [7,8,9]. It is progressive and fatal if untreated. The precursor LCs are secreted by an aberrant clonal population of B cells, most commonly bone marrow plasma cells. These cells secrete a single LC sequence, referred to as monoclonal to distinguish it from the polyclonal immune repertoire. The LCs are subunits of mature antibodies, which have undergone V(D)J recombination and somatic hypermutation for antigen selection, so that each individual with AL amyloidosis has an essentially unique amyloid precursor protein [8]. LCs may circulate as a complete antibody or as a free LC, without a heavy chain partner [10]. Only a minority of clonal LCs cause clinically relevant amyloid deposition, whereas most are efficiently removed from circulation. In multiple myeloma (MM), monoclonal LCs circulate at elevated concentrations but generally do not cause symptomatic amyloid deposition [11,12]. Therefore, the unique sequence of each LC—including its PTMs—appears to determine whether it will form amyloid fibrils and lead to AL amyloidosis.

Humans have two LC isotypes, kappa (κ) and lambda (λ), one of which is selected within each B cell clone. The *IGK* and *IGL* genomic loci, which encode κ and λ LCs, respectively, comprise multiple variable (*IGV_L_*), joining (*IGJ_L_*) and constant (*IGC_L_*) precursor germline gene fragments that are recombined to produce a functional gene. LCs derived from a subset of precursor genes are observed in patients with AL amyloidosis more frequently than in the polyclonal repertoire or MM [13,14]. Over 60% of AL clones express a LC derived from the *IGKV1-33*, *IGLV1-44*, *IGLV2-14*, *IGLV3-1* and *IGLV6-57* genes [14,15]. The precursor gene fragments that are used account for much of the sequence variation between LCs and can be thought of as reference sequences that are then further modified by somatic hypermutation. The protein sequence defines the 3-dimensional structure of the LC and determines which residues are co- or post-translationally N-glycosylated.

Antibody structures are formed from arrays of immunoglobulin domains, comprising two ß-sheets linked by a disulfide bond [8]. LCs have an N-terminal variable (V_L_) domain, involved in antigen recognition, and a C-terminal constant (C_L_) domain, that forms a tight interface with the heavy chain, including an inter-chain disulfide bond. V_L_-domains are encoded by germline *IGV_L_-IGJ_L_* gene rearrangements. Here, we describe each LC as being derived from a specific *IGV_L_* gene, either *IGKV* or *IGLV*. Within V_L_-domains, three complementarity determining regions (CDR 1, 2 and 3) with highly diverse sequences form loops that define the antigen binding site of most antibodies. These loops are scaffolded by framework regions (FR 1, 2, 3 and 4), which are more conserved between antibodies. In a prototypic antibody, two heavy chains and two LCs assemble into a heterotetramer with two antigen binding arms (known as a Fab for “fragment antigen binding”) and an effector region (Fc, for “fragment crystallizable”). Each Fab comprises the heavy chain variable and constant 1 domains (V_H_ and C_H_1) and the full-length LC. Free LCs may circulate as monomers or form homodimers, which can be stabilized by a disulfide bond between C_L_-domains.

Multiple studies have shown that AL amyloid fibrils are formed from LC-derived peptides in non-native cross-β conformations that have no structural homology with the native, β-sandwich immunoglobulin fold [4,5,16,17,18,19,20]. Therefore, unfolding of the native state after secretion from plasma cells appears to be required for aggregation. LC sequence features may drive amyloid formation by disfavoring the native LC structure, relative to that of the fibril and unstructured intermediate states [21]. Several examples of destabilizing residue changes in amyloidogenic, relative to non-amyloidogenic LCs, have been described [22,23,24].

Although a role for destabilizing mutations in LC amyloidogenesis is well established, the effects of PTMs are less clear. Multiple modifications have been observed in LCs from patients, both within amyloid fibrils and in the circulating precursor LCs. Proteolytic cleavage of parts of the C_L_-domain, which has a stabilizing effect on LCs [25,26], is almost ubiquitous, although it is not clear whether this occurs before or after aggregation [25,27,28,29,30]. The internal disulfide bond that stabilizes all immunoglobulin domains is retained within the structure of the fibrils, although the orientation of the peptide chains is inverted around the disulfide [17]. Cyclization of N-terminal glutamine residues to form pyroglutamate [4] and oxidative modifications of residues including cysteine and methionine [31,32,33] are also observed. N-glycosylation of asparagine residues, which has been observed in many AL-associated LCs [32,34,35,36,37,38], has gained a new prominence recently due to several studies showing that κ LCs from individuals with AL amyloidosis are more likely to be N-glycosylated than κ LCs associated with MM [39,40,41,42,43]. Notably, two high-resolution structures of λ AL amyloid fibrils are N-glycosylated, with electron density for the glycan visible in the maps [4,5]. These observations have led to the hypothesis that N-glycosylation is a biomarker for potentially amyloidogenic LCs, which could aid early diagnosis. However, the reasons for the association of AL amyloidosis with glycosylated LCs are unknown.

All human antibodies are N-glycosylated in their Fc regions [44]. In addition, around 10% of antibodies are N-glycosylated within their antigen-binding regions, apparently for functional reasons [45,46]. N-glycosylation is carried out co- or post-translationally by the oligosaccharyl transferase complex in the endoplasmic reticulum (ER) [10,47]. Most secreted or cell-surface proteins are N-glycosylated at asparagine residues within a short sequence motif or “sequon”, NxS/T, where N is asparagine, x is any residue other than proline and S/T is serine or threonine. Most sequons appear to be introduced into *IGV_L_-IGJ_L_* genes by somatic hypermutation [40,46].

Attachment of glycans alters the structure and function of glycoproteins [48]. Sugars are hydrophilic and resistant to burial within the hydrophobic cores of proteins, so glycans generally decorate the surface of glycoproteins. Glycans that have been observed on amyloid fibril structures are also exposed to the solvent [4,5,49]. Glycans prevent access to regions of the surface by some potential partners, while providing interaction sites for carbohydrate-binding proteins such as lectins [44,48]. Glycosylation may stabilize or destabilize the protein structure against unfolding, depending on the interactions that the glycan makes with surface residues [50,51,52]. Glycans may also increase protein solubility, by preventing self-association of hydrophobic surfaces [53]. Notably, natural N-glycosylation of antibody Fabs has been observed to increase thermal stability [54]. Based on these features, it seems reasonable to hypothesize that N-glycosylation generally reduces the propensity of proteins to aggregate.

Why then, is LC N-glycosylation associated with AL amyloidosis? Here, to address this question, we compare the fraction of LCs containing N-glycosylation sequons and the distribution of sequons within LC sequences between AL amyloidosis, MM and the polyclonal repertoire. Monoclonal, disease-associated LCs are taken from AL-Base, the Boston University database of LC sequences associated with AL amyloidosis and other plasma cell dyscrasias [13]. AL-Base has been updated with new sequences and an improved website, available at http://albase.bumc.bu.edu/aldb (accessed on 13 November 2024). These changes and further analysis of the sequences are described elsewhere [15]. As an additional control, we examine polyclonal sequences from the Observed Antibody Space (OAS) resource [55,56]. This large number of sequences allows the association between AL amyloidosis and N-glycosylated LCs to be explored in more detail than has previously been possible.

## 2. Results

### 2.1. N-Glycosylation Sequons Are Enriched in AL-Associated κ LCs

To ask whether LC V_L_-domain N-glycosylation is associated with amyloidosis, we counted the V_L_-domain sequences from AL-Base and OAS that harbor an NxS/T sequon (Table 1 and Figure 1). Three groups of LCs were analyzed: AL- and MM-associated LCs from AL-Base, corresponding to the AL and MM subcategories; and polyclonal LCs with unique sequences compiled from OAS [15]. For brevity, we refer to these as AL, MM and OAS LCs, respectively, in the following analysis. Only sequences with complete V_L_-domain sequences were analyzed. C_L_-domains were not analyzed, because only a minority of LCs have complete C_L_-domain sequences available. We did not attempt to predict which sequons are glycosylated [57], since bioinformatic prediction tools are designed to analyze arbitrary sequences, rather than antibody domains with known structural parameters [58].

Among 684 AL LCs with complete V_L_-domain amino acid sequences, 106 (15%) contain at least one sequon. A single AL LC harbors two sequons. For MM LCs with complete V_L_-domain amino acid sequences, 100 out of 969 LCs contain a sequon. Figure 1 shows the proportion of sequences containing a sequon, as a function of locus and origin. A significantly greater fraction of AL LCs harbors sequons, compared to MM LCs, with an odds ratio (OR) of 1.59 and 95% confidence interval (CI) of 1.34–3.41. Similarly, sequons are more frequent in AL LCs than in OAS LCs (OR 2.15; CI 1.24–5.67) (Figure 1A). Sequons are highly enriched in AL κ LCs compared to MM κ LCs (OR = 6.19; CI 1.50–57.9) and OAS κ LCs (OR = 7.82; CI 1.37–83.6) (Figure 1B). The association between AL amyloidosis and sequons is stronger for κ LCs than for all LCs because λ LCs are almost three times more frequent than κ LCs among AL clones, yet less likely to harbor a sequon.

AL-Base sequences were compiled from diverse studies, which may influence these results. Among the 243 AL LC sequences that have been identified at Boston University and deposited in AL-Base [13,60], 44 (18%) harbor sequons within their V_L_-domains. The correspondence between results from a single center and the larger database suggests that these data are applicable to the wider population.

### 2.2. Sequons Are Associated with Increased Proportions of Mutated Residues

Sequons in antibodies are thought to be created by somatic hypermutation [40,46]. Accordingly, antibodies harboring sequons in their variable domains have been observed to have accumulated more mutations, relative to their assigned germlines, than antibodies without sequons [46]. To test whether this is also the case for monoclonal LCs, we calculated the fraction of mutated residues for AL and MM LCs with and without a sequon (Figure 2). AL sequences harboring sequons have a significantly greater fraction of mutated residues than other AL sequences. Comparison of the median proportion of mutations per LC showed that κ and λ LCs with sequons had 1.9% and 1.8% more mutations than those without sequons, respectively. For MM LCs, a similar pattern was observed, but the differences were not significant: κ and λ LCs with sequons had 0.6% and 1.2% more mutations than those without sequons, respectively.

### 2.3. AL LCs Derived from a Subset of Precursor Genes Are Enriched in Sequons

We next asked whether N-glycosylation sequons were enriched among LCs derived from individual precursor germline genes (Figure 3). The number (Figure 3A) and proportion (Figure 3B) of sequons observed in LCs derived from each gene varies substantially. AL LCs derived from *IGKV1-16* are the most likely to contain sequons, in 15 out of 21 sequences (71%). A significantly higher fraction of AL LCs than MM LCs derived from the *IGKV1-33* (OR 6.59; CI 2.72–16.0) and *IGKV1-39* (OR 10.5; CI 2.16–51.3) germline genes harbor sequons (Figure 3C). AL LCs derived from *IGKV1-16*, *IGKV4-1*, *IGLV1-51* and *IGLV2-14* are also more likely to have sequons than MM LCs derived from these genes, but the comparisons are not significant after correction for multiple testing. Sequons are more frequently observed in AL LCs than OAS LCs derived from five genes, *IGKV1-33* (OR 5.17; CI 3.06–8.80), *IGKV1-39* (OR 5.38; CI 2.50–11.6), *IGKV1-16* (OR 5.07; CI 1.98–13.0), *IGLV1-51* (OR 2.76; CI 1.19–6.37) and *IGLV2-14* (OR 2.56; CI 1.46–4.48) (Figure 3C). Notably, AL LCs derived from the *IGLV6-57* gene are significantly less likely to harbor a sequon than OAS LCs derived from this gene (OR 0.20; CI 0.051–0.78) (Figure 3C). *IGLV6-57* is the most frequently observed precursor gene in AL amyloidosis [14,15]. The comparison between *IGLV6-57*-derived AL and MM LCs did not reach significance, likely because only nine MM LCs (Figure 3A) are derived from this gene.

Three *IGV_L_* precursor germline genes encode NxS/T sequons, *IGKV5-2*, *IGLV3-22* and *IGLV5-37*. No AL LCs in AL-Base are derived from these genes. Sequons are present in four of the five MM LCs derived from *IGKV5-2* and two of the three MM LCs derived from *IGLV5-37*.

### 2.4. Sequons in AL and MM LCs Occur in Similar Positions

Sequons are distributed unevenly within V_L_-domain sequences [46]. We aligned each sequence to the International ImMunoGeneTics (IMGT) Information System numbering, which has a total of 127 possible positions for residues, including gaps to accommodate insertions [61]. Residues at the same IMGT position are in comparable positions in the 3D structures of immunoglobulin domains, which allows residues to be compared between LCs derived from different germline genes [61]. Note that CDR3 varies in length between LCs and is structurally heterogeneous, so the location of sequons in this region is less well defined by this analysis than that of other sequons.

We considered κ and λ LC separately and only analyzed the first sequon in each V_L_-domain. Sequons occur at 14 positions in κ AL LCs and 14 positions in λ AL LCs, of which six positions occur in both types (Figure 4). No AL LCs have germline-encoded sequons. Sequons primarily occur at the sites of “progenitor” N-glycosylation sequences [46] in the respective germline gene (Figure 4, orange bars). These are sequences where only a single nucleotide change is required to create a sequon in the translated protein [46]. The distribution of sequon positions is different between the two types of LC: 56 out of 67 κ sequons (84%) are within FR3, whereas 20 out of 38 λ sequons (53%) are within CDR3. Residues 75, 86 and 88 are the most frequent locations of sequons among κ LCs. Residues 109, 110 and 113 are the most frequent locations of sequons among λ LCs. The Pearson correlation coefficient between sequon counts at each position for κ AL LCs and λ AL LCs is 0.076 (*p* = 0.4), indicating no correlation between the two isotypes.

The locations of sequons in MM LCs are similar to those of AL LCs. Sequons were identified at 15 positions in κ MM LCs and 14 positions in λ MM LCs, four of which occur in both types (Figure 4). Nine sequon positions in κ LCs and 10 sequon positions in λ LCs are observed in both AL and MM LCs. Position 86 is the most frequently observed sequon position in both AL and MM κ LCs. The Pearson correlation coefficients between AL and MM sequon distributions are 0.72 (*p* = 1.5 × 10^−19^) and 0.83 (*p* = 4.8 × 10^−32^) for κ and λ LCs, respectively. This indicates that distributions of sequons are more similar between LCs associated with AL and MM than between κ and λ LCs.

Sequons are observed at many more positions in OAS LCs than in AL or MM LCs, consistent with the much greater number of sequences analyzed. Of the 125 IMGT positions where a sequon could be detected (see methods), 104 κ positions and 106 λ positions have sequons in at least one OAS LC. All sequon positions in AL and MM LCs have equivalents in OAS LCs. The positions of AL sequons that correspond to the least common OAS sequons are position 18 for κ LCs and 37 for λ LCs, for which 896 and 893 OAS sequons are observed, respectively. The most frequent sequon position in κ AL LCs, 86, is the second most frequent sequon position in OAS κ LCs. For AL λ LCs, the most frequent sequon position, 109, is the sixth most common sequon position in OAS λ LCs. Therefore, we conclude that all sequons in AL LCs have equivalents in the polyclonal repertoire.

Sequons were previously reported to be enriched in IMGT strands D and E of FR3 within AL κ LCs [40]. The fraction of sequons positioned in each IMGT-defined structural element is shown in Figure 5. Note that the IMGT “strands” do not correspond exactly to secondary structure elements. Sequons in strand E of κ LCs account for 57%, 36% and 23% of AL, MM and OAS LCs, respectively. However, a subset of *IGKV* genes is over-represented in AL versus MM and OAS LCs [15]. Among LCs derived from the *IGKV1* family, which account for 126 of 160 AL κ LCs (78.8%) and 54 of 67 AL κ sequons (80.6%), strand E accounts for 57%, 45% and 17% of AL, MM and OAS sequons, respectively. In contrast, only three AL LCs with sequons are derived from *IGKV3*-family genes, and strand E is the location of 33%, 18% and 25% of AL, MM and OAS sequons, respectively. Therefore, the enrichment of FR3 sequons in AL amyloidosis may be due to the over-representation of a subset of precursor genes among AL clones.

### 2.5. Sequons Occur in Similar Sequence Contexts

The residues around a sequon alter its propensity to be glycosylated in the ER [6,57,62]. Alignments of the residues around each AL and MM sequon are shown as sequence logos [63] in Figure 6. Residues surrounding the sequons vary, but are similar between AL and MM LCs within each isotype. AL LCs are more likely than MM LCs to have threonine in the +2 position, which is associated with a higher frequency of N-glycoslyation [57]. For κ LCs, 39 of 68 AL sequons (57%) include a threonine residue compared to 27 of 62 MM sequons (44%). For λ LCs, 22 of 39 AL sequons (56%) and 22 of 38 MM sequons (58%) include a threonine residue.

### 2.6. NxC Sequons Occur in LC CDRs

Glycosylation has been reported at sequon-like motifs where a cysteine residue replaces the serine or threonine, denoted NxC [23]. Although it is not clear whether such motifs would be glycosylated in vivo, we identified five NxC sequons within AL LCs and one within a single MM LC (Table 2). All NxC sequons occur within CDR regions.

### 2.7. Sequon Positions Are Compatible with Native Antibody Structures

We next asked whether the positions of sequons within the 3-dimensional structures of LC V_L_-domains could help to explain their association with AL amyloidosis. Glycosylation is expected to occur on surface residues with solvent-exposed sidechains in order to minimize disruption to the folded core of the V_L_-domain. Sequons in positions that would otherwise be buried could lead to destabilized LCs, which are associated with amyloidosis [22]. LC surface and core residues were identified from crystal structures of germline LC V_L_-domains. We used PDB entries 2Q20 [64] and 6SM1 [65], which are isolated κ *IGKV1-33* and λ *IGLV2-14* V_L_-domains with germline-identical sequences, as representative examples for κ and λ LCs, respectively. For each isotype, IMGT positions were defined as being on the surface, buried within the hydrophobic core or as residues that form the conserved interface with heavy chains [40,41]. These residues are shown for the two V_L_-domains in Figure 7.

All sequon positions from AL LCs correspond to residues whose sidechain is solvent-exposed in the structure of an isolated V_L_-domain (Figure 8 and Figure 9). A similar pattern is observed for MM and OAS LCs. Two MM sequons were observed at position 108 in λ LCs, which is the only potentially N-glycosylated residue that is predicted to be buried in the 6SM1 structure that was analyzed. However, this CDR3 residue may be solvent-exposed in other LCs.

All positions are exposed on the surface of the V_L_-domain and are accessible to the solvent, consistent with them being available for N-glycosylation. However, residues in CDR3 form part of the interface between the protein domains of both types of dimers, so sequons at these positions could alter the LC quaternary structure. The three most frequent sequon positions for AL λ LCs, IMGT positions 109, 110 and 113, are within CDR3 and make contact with residues in the other chain of the dimers (Figure 9).

### 2.8. Sequon Positions Differ in Their Environment Among AL Fibril Structures

Finally, we asked whether the positions of sequons within AL LCs are compatible with the known structures of AL amyloid fibrils. High resolution fibril structures have been determined for seven distinct clonal λ LCs [46]. There are no published structures of κ AL amyloid fibrils. For each λ AL fibril structure, we identified residues that are glycosylated in either κ or λ LCs.

Figure 10 shows the number of sequons observed for each IMGT residue position within κ LCs. Position 86, which is the most frequent site of AL κ sequons (dark blue spheres in Figure 10), is resolved in all seven structures along with positions 88 and 90 within FR3. In the 6HUD and 8R47 fibrils, position 86 is on the surface and exposed to the solvent, so N-glycosylation would not disrupt the fibril structure. This position is buried within the core of the 6IC3, 9EME and 9FAA fibrils, and therefore incompatible with N-glycosylation. In the 6Z1O and 7NSL fibrils, position 86 is close to unresolved regions of the fibril but its positions does not appear to be compatible with N-glycosylation due to the concave surface of the fibrils in these regions.

The locations of λ sequons also vary between fibril structures (Figure 11). Of the 14 positions at which sequons are detected among AL λ LCs, none are solvent-exposed in all seven fibril structures. The most frequent region in which λ sequons are observed, CDR3, is within the structured region for all seven fibrils. Position 109, the most common location of sequons, is buried within the core of the 6Z1O, 8R47, 9EME and 9FAA fibrils (dark blue spheres in Figure 10). This position is exposed on the surface of the 6HUD, 6IC3 and 7NSL fibrils.

## 3. Discussion

Our comparative analysis of monoclonal AL LCs, monoclonal MM LCs and polyclonal OAS LCs is consistent with previous observations that N-glycosylation sequons are significantly enriched among AL κ LCs, compared to non-AL κ LCs [39,41,42] (Figure 1). For λ LCs, sequons occur at similar rates between AL and non-AL LCs. LCs harboring sequons have more mutations than other LCs (Figure 2). Sequons are significantly more frequent (FDR ≤ 0.05) in AL LC than in OAS LCs among five precursor genes: *IGKV1-16*, *IGKV1-33* and *IGKV1-39*; and *IGLV1-51* and *IGLV2-14* (Figure 3). The locations of sequons within LC sequences are similar within LCs of the same isotype, predominantly occurring at progenitor N-glycosylation sites where only a single nucleotide change is required to create an NxS/T sequon [43,65] (Figure 4 and Figure 5). The residues around AL and MM sequons are similar, although AL LCs more frequently have threonine in the +2 position, consistent with efficient N-glycosylation [57] (Figure 6). Among AL LCs, sequons are invariably located at residues that would be exposed to the solvent in an isolated variable domain, but the majority of λ sequons occur in residues that are close to the interface with the LC partner in homodimers and the heavy chain in antibody Fab complexes (Figure 8 and Figure 9). No sequon positions are located on the surface of all seven known λ amyloid fibril structures (Figure 10 and Figure 11).

Glycosylation has been suggested as a biomarker for amyloidosis risk [40]. It is possible to identify glycosylated monoclonal free LCs from blood or urine without determining their sequence, using biochemical analysis or mass spectrometry [48]. Such a test would not require LC sequence information and could be implemented using existing technology. However, although sequons are highly enriched among AL κ LCs, most AL amyloidosis clones express λ LCs (Table 1). The fraction of κ LCs with a sequon is 67/684 (9.8%) in AL LCs and 62/969 (6.4%) in MM LCs. Because AL amyloidosis is much less prevalent than MM [66,67], the presence of a glycan or sequon cannot, in itself, provide sufficient information to contribute to diagnosis. Further work is needed to understand the contexts in which N-glycosylation promotes amyloidosis.

The context of N-glycosylation may be one factor that determines its effect on LCs. Sequons are more frequent in AL LCs derived from a subset of precursor genes (Figure 3 and Figure 5). Therefore, the enrichment of sequons among AL κ LCs is not simply due to the over-representation of precursor genes which are frequently glycosylated. Of note, AL LCs derived from *IGLV6-57*, which is the gene most strongly associated with AL amyloidosis, are significantly less likely to harbor sequons than *IGLV6-57*-derived LCs from the polyclonal repertoire (Figure 3C). Nevone and coworkers identified glycosylation within FR3 as a predictor of amyloidosis risk [67,68]. Our data are consistent with this hypothesis. However, FR3 sequons are enriched in genes that are observed more frequently in AL amyloidosis than in MM (Figure 5). Therefore, the over-representation of FR3 sequons is related to the over-representation of LCs which are frequently glycosylated at these positions. It is not clear whether frequent N-glycosylation on FR3 is a cause or a consequence of the association between *IGKV1* genes and AL amyloidosis.

N-glycosylation can increase protein stability and solubility, and burial of hydrophilic glycan moieties within the hydrophobic cores of proteins is thermodynamically unfavorable [68]. The locations of sequons within LCs are not consistent with substantial disruption of the variable domain structure that would lead to increased unfolding-linked aggregation (Figure 8 and Figure 9). Although mutation to asparagine of a residue that would otherwise be buried within the context of a sequon is possible, the low frequency of these events is consistent with the hypothesis that such structures would not fold sufficiently well for surface expression on B cells and subsequent clonal selection. Glycosylation of CDR3 residues may disrupt the interface between the LC and its heavy chain partner sufficiently that less antibody, and more free light chain, is released into circulation. Moreover, CDR3 glycans may also disrupt the homodimerization of free LCs, since two glycans in close proximity may be energetically unfavorable; that is, the reduction in the available space for each glycan carries an entropic cost that could destabilize the homodimer. Dimerization of free LCs can be protective against aggregation, an effect that is exploited by small molecules that stabilize LCs against unfolding and proteolysis [69]. Therefore, disruption of such dimers could favor misfolding. We hypothesize that this disruption of LC homodimers by CDR3-linked glycans is a factor in amyloidosis of N-glycosylated λ LCs.

Misfolded glycoproteins are triaged by the calnexin/calreticulin chaperone system within the ER and subject to ER-associated degradation [7]. N-glycosylation may allow unstable LCs that would otherwise misfold within the cell to be exported, where they can go on to aggregate elsewhere. This is analogous to the situation in ATTR amyloidosis, where highly destabilized transthyretin variants are not exported from the liver due to stringent quality control within hepatocytes [69]. Secretion of these LCs would require additional cellular resources, both to synthesize the glycans and to allow their refolding by ATP-dependent chaperones within the ER and, when necessary, proteasomal degradation [70]. Such resources may not be compatible with the more rapid proliferation of MM plasma cells, but may be tolerated by slow-growing AL plasma cells. This mechanism implies that glycosylated LCs are less stable than other LCs, such that they are exported less efficiently without their glycan, which could be tested experimentally. Glycosylated LCs may also impose more load on cells’ proteostasis machinery, potentially making them more sensitive to proteasome inhibition, which is part of the standard of care for AL amyloidosis treatment [11,12].

Assuming that glycans must be solvated and displayed on the surface of fibrils, each sequon position is compatible with only a subset of known AL fibril structures (Figure 10 and Figure 11). We anticipate that when structures of glycosylated κ LCs are solved, they will form new amyloid folds. The presence of glycans on fibrils may promote or prevent interactions with other factors in order to enhance the stability of fibrils. For example, glycans may prevent access to fibrils by proteases, or serve as signals that deter phagocytic cells. It is possible that glycosylation could prevent binding of therapeutic antibodies that are undergoing clinical trials as amyloid-depleting therapies [13,15], so fibrils’ glycosylation status could be an important marker for the use of such therapies.

The major limitation of this study is the relatively small number of monoclonal LC sequences that contain sequons, which leads to wide confidence intervals on the calculated odds ratios, reducing confidence in the analysis. This limitation is compounded by the uncertainty of which sequons are actually N-glycosylated in vivo [40,57,71]. Identifying which sequon asparagine residues are modified in patient LCs will be important to validate this work. Like other studies of LCs, the classification of clinical cases as amyloidosis may underestimate the true frequency of amyloid deposition [55,56,70,71,72,73,74,75]. A fraction of MM cases involves amyloid that may not be considered clinically significant [55,56,71,72,73,74,75,76]. Similarly, OAS LCs may be amyloidogenic in the context of a plasma cell dyscrasia. However, because AL amyloidosis is rare, it is reasonable to suppose that most LCs are resistant to amyloidogenesis, even as monoclonal proteins. Another limitation is that we only consider the V_L_-domains of LCs, although N-glycosylation may also occur within the C_L_-domain. Finally, our structural analysis relies on the correspondence of IMGT numbering with V_L_-domain structure, which may not be true for all LCs.

In summary, our analysis has identified several previously unreported features of glycosylation among AL LCs. Although there is a strong association between glycosylation and AL for a subset of precursor genes, these LCs represent only a minority of AL amyloidosis cases and the presence of glycosylation should not be considered diagnostic for amyloidosis. Glycosylation is clearly neither necessary nor sufficient for amyloidosis. We suggest several testable hypotheses that might further explain how glycosylated LCs can lead to amyloid disease. We anticipate that the characterization of more LCs, both as soluble proteins and amyloid fibrils, will allow these hypotheses to be tested in the near future.

## 4. Materials and Methods

### 4.1. LC Sequences

Sequences in AL-Base are assigned to a category and subcategory that describes whether a LC is associated with amyloid deposition, and the clinical diagnosis described in the original report [77]. AL-Base contains 684 and 969 LC sequences associated with AL amyloidosis and MM, respectively, for which a complete amino acid sequence of the V_L_-domain is available (Table 1). Sequences with missing or ambiguous residues were not included in the analysis. We excluded sequences where AL amyloidosis was diagnosed in the context of another malignancy, including MM, chronic lymphocytic leukemia and Waldenström macroglobulinemia, which correspond to the AL-Base subcategories AL/MM, AL/CLL and AL/WM, respectively. C_L_-domain sequences were not analyzed. AL-Base sequences were assigned to precursor germline genes using the IMGT HighVQuest and DomainGapAlign tools [78]. As a comparison, we identified 8,047,747 unique LC sequences from six repertoire sequencing studies of healthy individuals from the Observed Antibody Space (OAS) resource [55,56,72,73,74,75,76,77]. These sequences had previously been assigned to germline genes using the IgBLAST tool during deposition to OAS, so we did not attempt to reassign the sequences. Alleles of each germline gene were analyzed together. Paralogous *IGKV* genes, e.g., *IGKV1-16* and *IGKV1D-16*, from the proximal and distal loci, respectively, were considered to be identical [78].

All analysis of sequons was carried out on protein sequences. Nucleotide sequences were not available for all LCs and were not used. Data analysis was carried out in R v 4.2.2 [79] via the RStudio environment [80], primarily with the Tidyverse suite of tools (v 1.3.2) [81]. N-glycosylation sequons were identified and located within sequences using text recognition with the regular expression “N[^P][ST]”, implemented with the Unix shell *grep* utility or the stringr package [82] within R. This method cannot observe sequon asparagine residues in the final two positions of the V_L_-domain, so a total of 125 IMGT positions was available for analysis. The residues around each sequon were extracted and used to create sequence logos using the ggseqlogo package [83], where the asparagine residue was assigned to position 0. Logos were scaled by residue frequency, rather than information content.

All sequences were aligned to the IMGT numbering system using the ANARCI tool [84], which we found to be more consistent than other tools at positioning residues around CDR3. We excluded OAS sequences that could not be aligned via ANARCI and excluded sequons at residues that did not map to IMGT positions, which are located within large insertions in CDR loops. A total of 631,433 OAS LCs had sequons available for position analysis. The locations of progenitor N-glycosylation sites, which are germline sequences where only a single nucleotide change is needed to create a sequon, were taken from the list provided by van der Bovenkamp and coworkers [84].

We calculated the number of amino acid changes in each V_L_-domain as previously described [85]. Each LC protein sequence was aligned to its precursor germline *IGV_L_* and *IGJ_L_* genes using the Biostrings package in R [85] and the number of residue substitutions, insertions and deletions was calculated. The fraction of mutations was defined as:$$Fraction\;mutated=\frac{Substitutions}{Germline\;sequence\;length}+\frac{Insertions}{Germline\;sequence\;length+Insertions}+\frac{Deletions}{Germline\;sequence\;length+Deletions}$$

### 4.2. Statistics

The fraction of mutations in AL and MM LCs was compared using a non-parametric Wilcoxon rank-sum test. Significance was corrected for multiple testing using the Benjamini-Hochberg false discovery rate (FDR) method [65], where FDR ≤ 0.05 was considered significant.

The fraction of sequences harboring sequons was compared between groups of LCs using logistic regression, implemented via generalized linear models in R. Models were created with and without stratifying by locus and precursor gene. Odds ratios (OR) and 95% confidence intervals (CI) were calculated from the model coefficients, or calculated manually. For manual calculations, an adjustment factor of 0.1 was added to all counts to avoid division by zero. We compared sequon fractions among all precursor genes, but only results from those genes where at least 10 AL LC sequences were available are reported. FDR ≤ 0.05 was considered significant.

For comparisons between sequon positions, the Pearson correlation coefficient *r* was calculated across the 125 IMGT positions where sequons could potentially be detected, including those where no sequon was observed.

### 4.3. Structural Analysis

All LC sequences harboring sequons were used to investigate the positions of sequons within 3D structures of folded and fibrillar LCs. In cases where more than one sequon was present in a LC, only the most N-terminal sequon was considered.

LC residues were assigned to the surface, core or interface regions of the 3D structures using ChimeraX [86]. Residues with less than 10% surface exposure, relative to reference peptides [87], were examined manually and all residues were assigned to either the core or surface of the molecule. Interface residues were defined as the minimum set of LC residues that form contacts with the heavy chain in multiple antibody structures [88], corresponding to IMGT positions 42, 44, 49, 52, 55, 103 and 118. Sequon positions were mapped on to the structures of V_L_-domain homodimers, antibody Fabs and amyloid fibrils using ChimeraX. The IMGT residue numbering for each structure was determined using ANARCI [84]. The packing of V_L_-domains in the crystal structures 2Q20 and 6SM1 is similar to that observed in full-length LC crystal structures, so these structures were used as models of κ and λ LC homodimers, respectively. Antibody structures 8X0X [88] and 7CZW [89] were identified as examples of Fabs with LCs derived from *IGLV2-14* and *IGKV1-33*, respectively. Both Fabs are from functional human antibodies against SARS-CoV2 Spike protein, isolated from COVID-19 patients.

Positions of AL-associated sequons were also mapped onto high resolution amyloid fibril structures reported for seven distinct clonal λ LCs, PDB entries 6HUD (*IGLV6-57*); 6IC3 (*IGLV1-44*); 6Z1O (*IGLV3-19*); 7NSL (*IGLV1-51*); 8R47 (*IGLV3-19*); 9EME (*IGLV3-19*) and 9FAA (*IGLV3-1*) [4,5,16,17,18,19]. There are no published high-resolution structures of κ LC amyloid fibrils, so both κ and λ sequons were mapped to the seven λ fibril structures. Sequons at IMGT positions that are not represented in the fibril PDB files were excluded.

## Figures and Tables

**Figure 1 pharmaceuticals-17-01542-f001:**
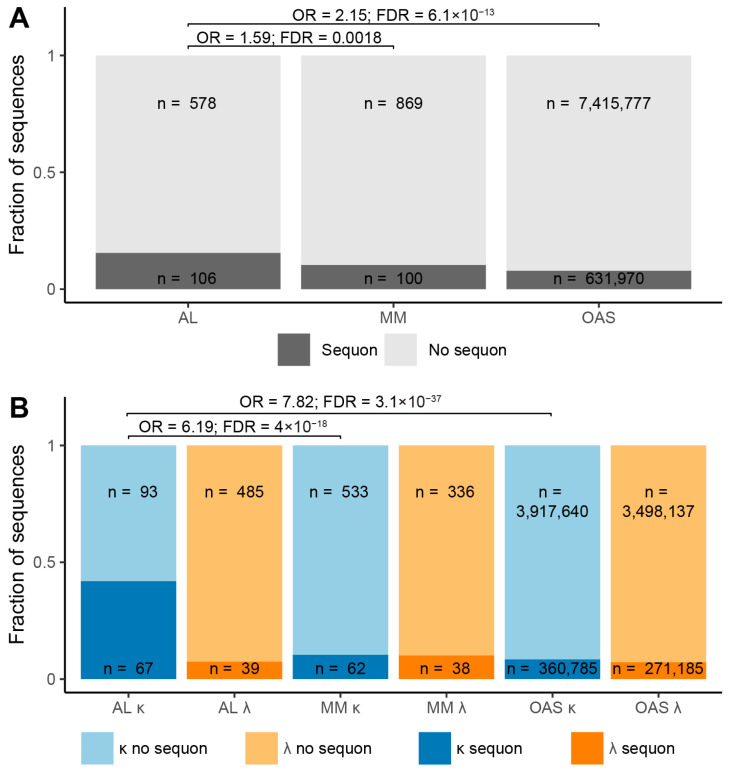
The fraction of sequences with NxS/T N-glycosylation sequons differs between AL amyloidosis, multiple myeloma (MM) and the polyclonal repertoire, represented by sequences from Observed Antibody Space (OAS). The proportion of LCs associated with AL or MM, or from the OAS repertoire, with or without an NxS/T sequon is shown in dark and light colors, respectively; κ LCs are blue and λ LCs are orange. Odds ratios (OR) for selected comparisons are shown. Significant differences are shown using false discovery rate (FDR) to correct for multiple comparisons [59]. (**A**) All LCs, with ORs for the AL vs. MM and AL vs. OAS comparisons. (**B**) LCs separated by isotype, with ORs for comparisons between κ groups.

**Figure 2 pharmaceuticals-17-01542-f002:**
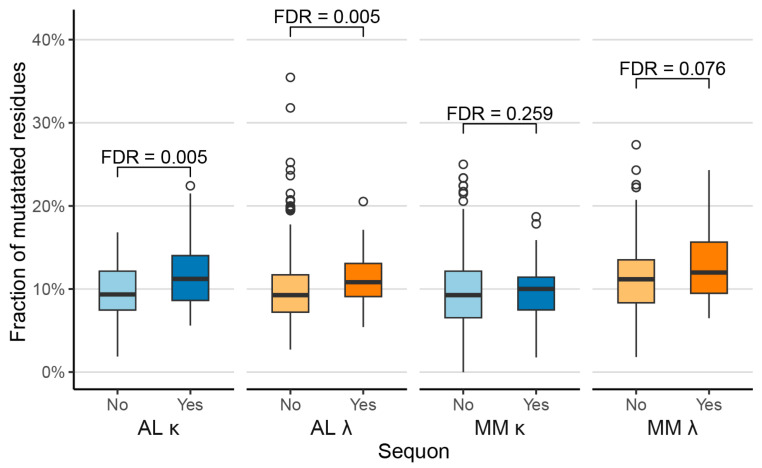
Glycosylation is associated with an increased number of somatic mutations. The number of amino acid residue substitutions, insertions and deletions in each LC V_L_-domain is shown as a percentage of its length. The box and whisker plots show median (central bars), inter-quartile range (boxes), distance to the non-outlier data (whiskers) and outlying points (circles). Blue and orange denote κ and λ LCs, respectively. Significance values, corrected for multiple testing, are shown for each comparison.

**Figure 3 pharmaceuticals-17-01542-f003:**
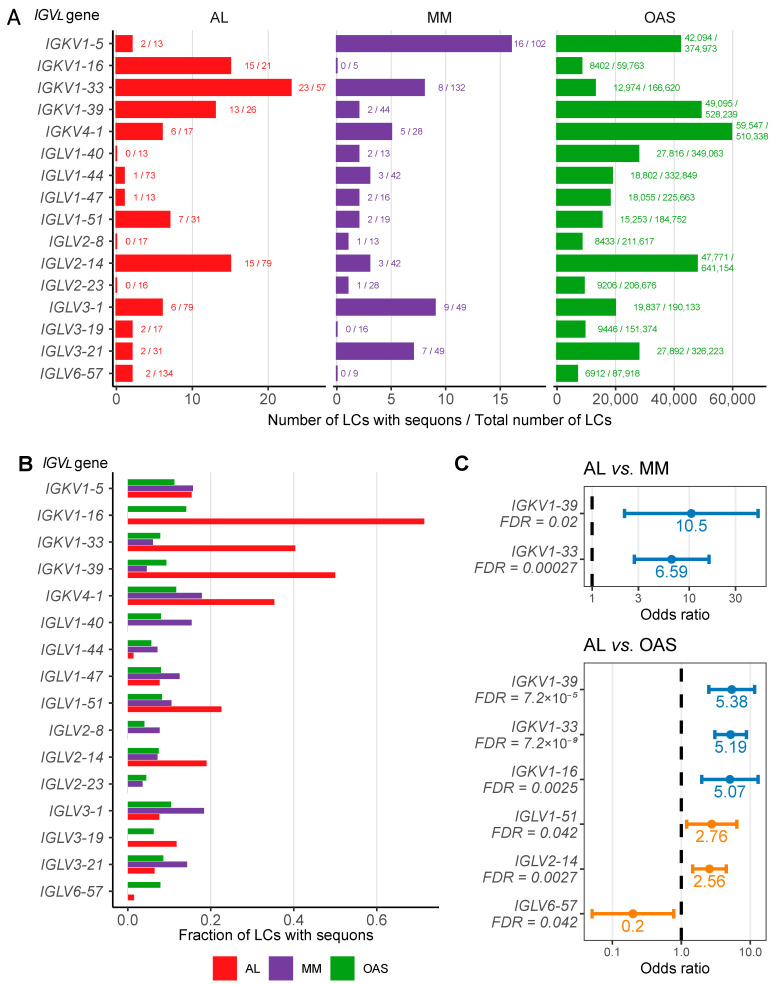
LCs derived from a subset of genes are significantly enriched in sequons. (**A**) Numbers of LCs derived from each germline gene that harbor an NxS/T sequon. Data are shown for genes from which at least 10 AL LCs are derived. Counts are shown as a fraction of the total number of LCs derived from that gene. AL, MM and OAS LCs are shown in red, purple and green, respectively. (**A**) The number of LCs associated with each gene for AL, MM and OAS LCs is shown as a fraction of the total number of sequences derived from that gene. (**B**) Fractions of LCs derived from each germline gene that harbor an NxS/T sequon. (**C**) Odds ratios and 95% confidence intervals for the relative frequency of sequons among LCs from different origins where a significant difference was observed (FDR ≤ 0.05). Blue and orange symbols show κ and λ LCs, respectively.

**Figure 4 pharmaceuticals-17-01542-f004:**
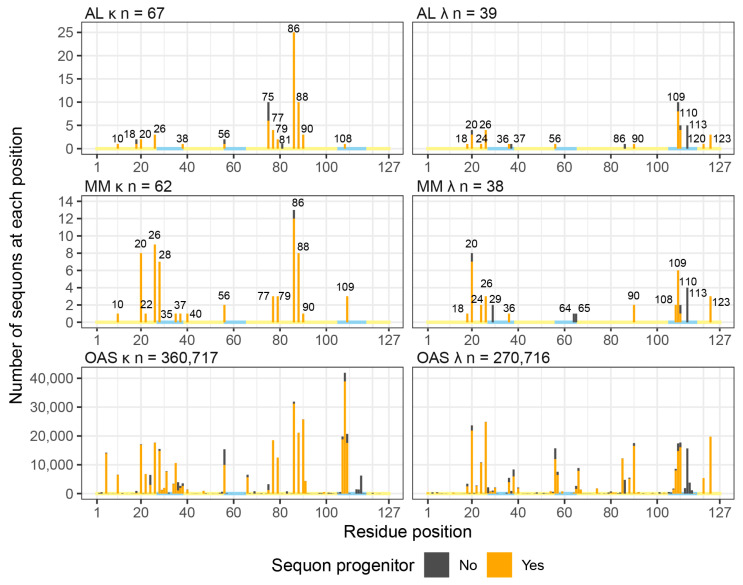
Positions of sequons within V_L_-domains. The number of sequons observed at each IMGT position is shown for κ and λ LCs associated with AL and MM, and from the polyclonal OAS repertoire. Orange and black bars show positions where a sequon progenitor is present and absent, respectively, in the assigned germline gene. Yellow and blue lines along the x-axes represent FR and CDR positions, respectively. Positions of sequons in AL and MM LCs are highlighted.

**Figure 5 pharmaceuticals-17-01542-f005:**
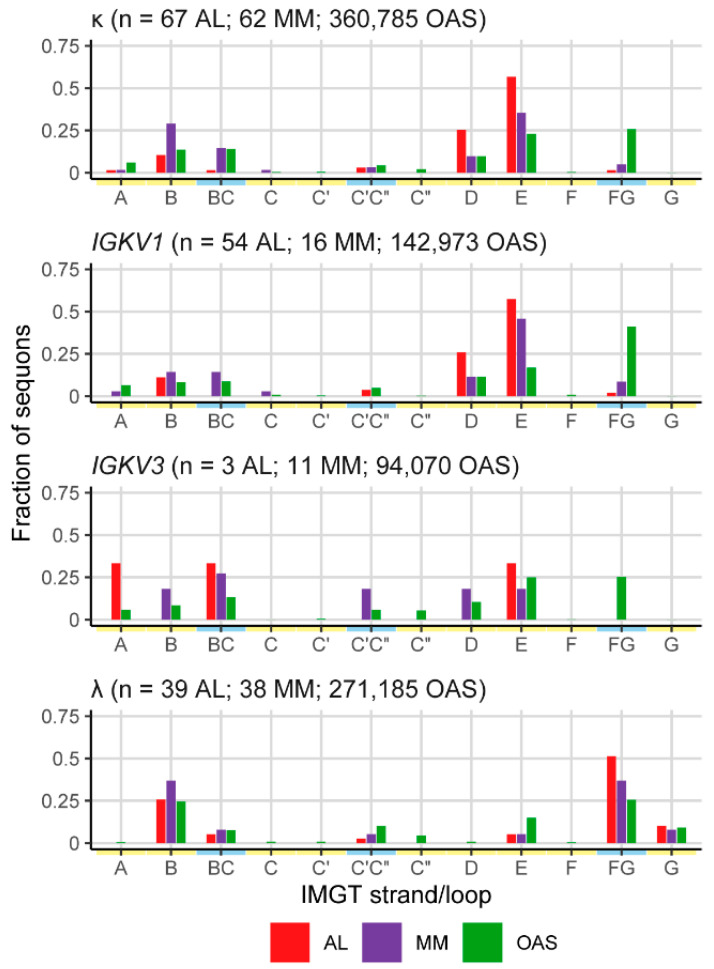
Location of sequons within LCs among IMGT-defined structural elements. AL, MM and OAS LCs are shown in red, purple and green, respectively. Yellow and blue lines along the x-axes represent FR and CDR positions, respectively.

**Figure 6 pharmaceuticals-17-01542-f006:**
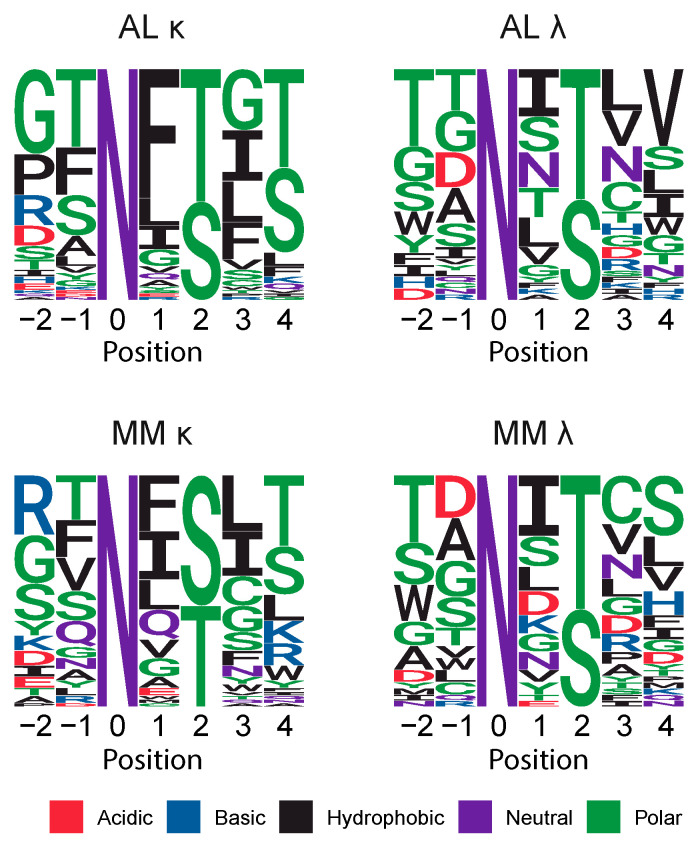
Residues around the sequon asparagine are similar between AL and MM LCs. Sequence logos showing the proportion of each residue observed around each sequon. Numbering is relative to the asparagine residue. Colors show the chemical properties of each residue.

**Figure 7 pharmaceuticals-17-01542-f007:**
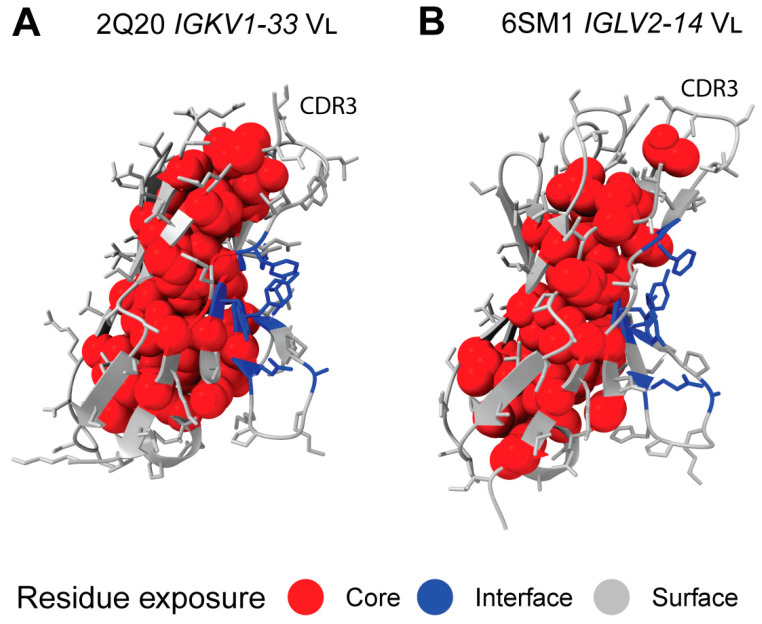
Surface exposure of residues in representative germline-identical V_L_-domains. Each domain is oriented so that the CDR3 residues (indicated in the figures as CDR3) are at the top right, and the C-terminal residues, which connect to the C_L_-domain, are at the bottom left. The protein backbone is shown in cartoon representation, with sidechains modeled as sticks for surface-exposed residues and spheres for residues buried in the hydrophobic core. Residues are colored according to solvent exposure: grey, surface-exposed; red, buried in the core; blue, interacting with the partner chain.

**Figure 8 pharmaceuticals-17-01542-f008:**
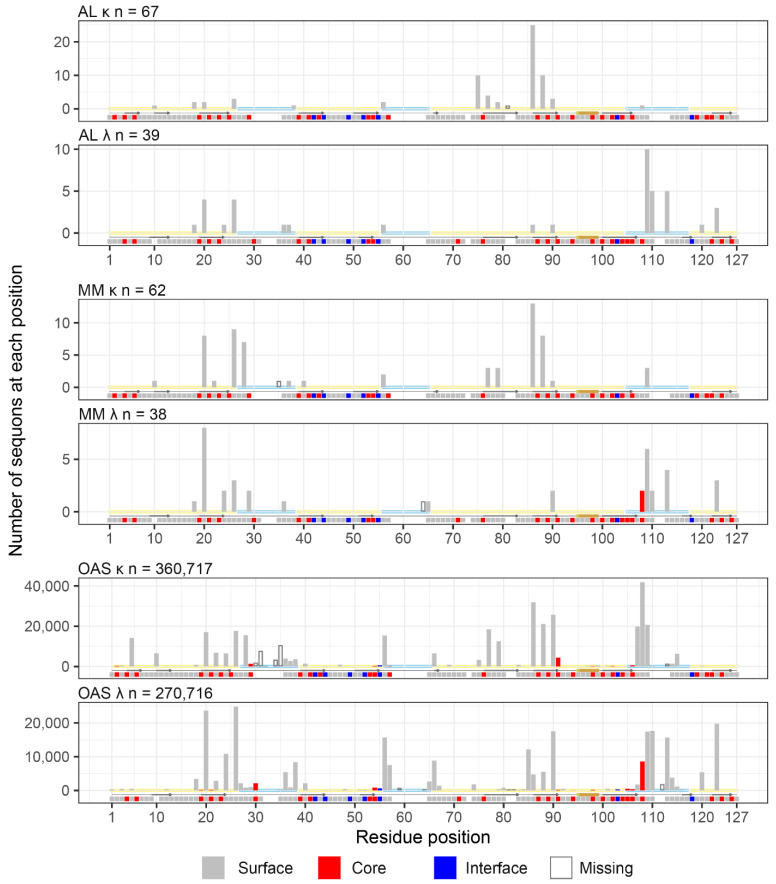
Most sequon positions are compatible with glycosylation. The number of sequons observed at each position is shown by bars, as for Figure 4. Solvent exposure is shown by the color of the bars and by squares under the plots. IMGT positions corresponding to residues on the surface of V_L_-domains, buried in the VL-domain core, and at the interface with heavy chains are shown in grey, red and blue, respectively. Solvent exposure was determined based on the structures of the isolated germline V_L_-domains for *IGKV1-33* (PDB entry 2Q20) and *IGLV2-14* (PDB entry 6MS1), which were used to represent κ and λ LCs, respectively. Hollow bars and gaps represent IMGT positions that are not represented in these structures. Yellow and blue lines along the x-axes represent FR and CDR positions, respectively. Arrows and cylinders below the axes show the positions of ß-sheets and α-helices defined in the structures.

**Figure 9 pharmaceuticals-17-01542-f009:**
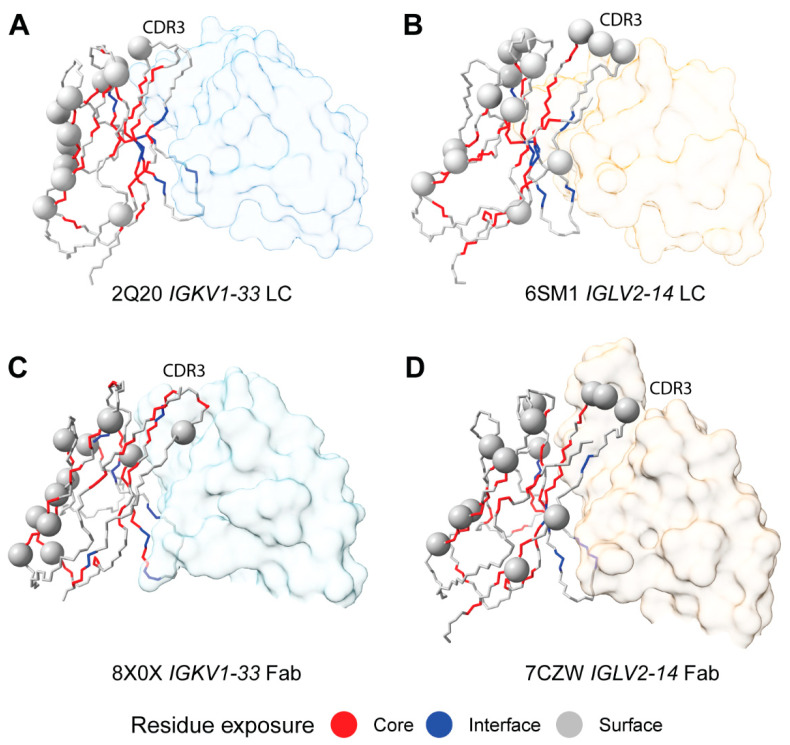
AL sequon positions mapped to LC structures in V_L_-domain homodimers (**A**,**B**) and antibody Fab complexes (**C**,**D**). Complexes are oriented so that the LC CDR3 residues (indicated in the figures as CDR3) are at the top and center, and the C-terminal residues, which connect to the C_L_-domain, are at the bottom left. One LC is shown as a backbone trace, with the Cα positions where sequons are observed shown as spheres. Residues are colored according to solvent exposure as for Figure 7: grey, surface-exposed; red, buried in the core; blue, interacting with the partner chain. The second LC of the dimer, and the heavy chain of each Fab is shown in surface representation.

**Figure 10 pharmaceuticals-17-01542-f010:**
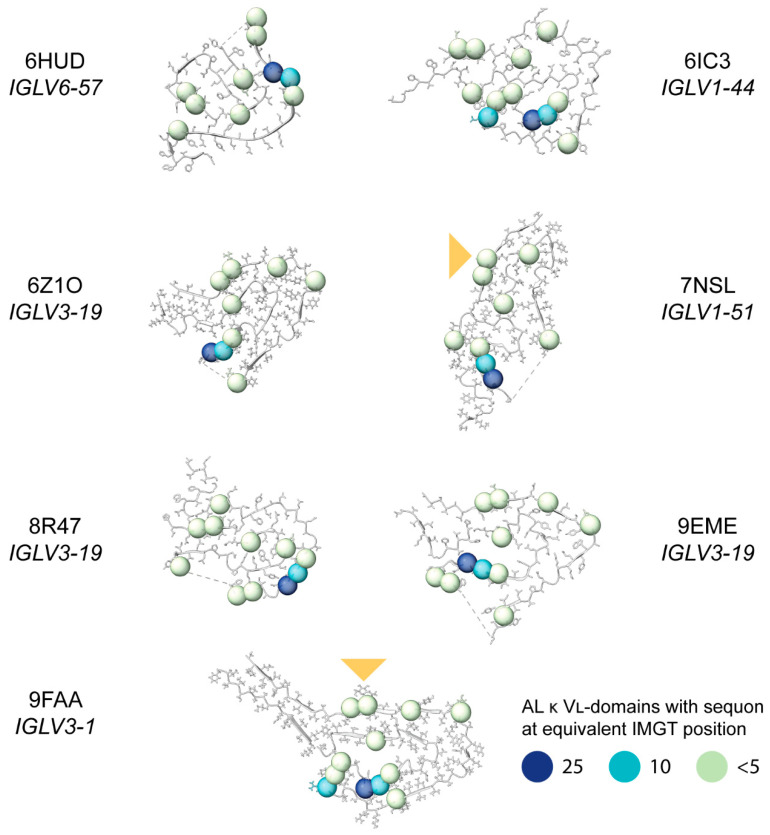
Positions of sequons observed in κ AL LCs, mapped onto the published structures of AL λ amyloid fibrils. The position of the glycans in the fibril structures of 7NSL and 9FAA are shown with a yellow triangle. All structures are oriented so that cysteine 23 is on the upper side of the disulfide and cysteine 104 is on the lower side. Dashed lines indicate missing density from the structures. Residue positions where sequons asparagine residues are observed in AL κ LCs are shown as spheres, colored to indicate the number of sequons at each position (data from Figure 4 and Figure 8). Note that not all IMGT positions are occupied in each structure, so not all the sequon positions can be shown.

**Figure 11 pharmaceuticals-17-01542-f011:**
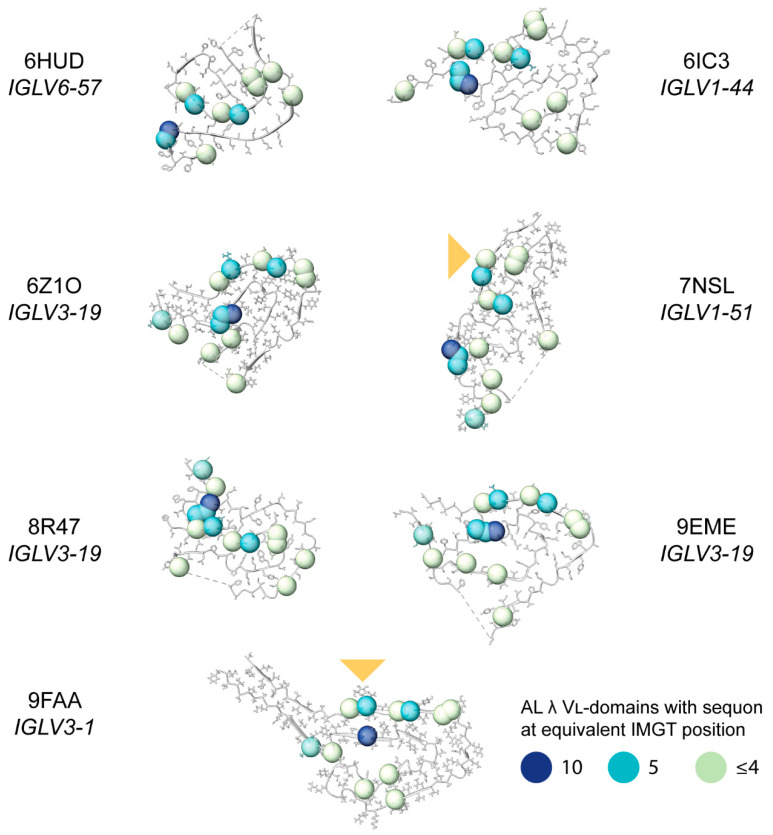
Positions of sequons observed in λ AL LCs, mapped onto the published structures of AL λ amyloid fibrils. The position of the glycans in the fibril structures of 7NSL and 9FAA are shown with a yellow triangle. All structures are oriented so that cysteine 23 is on the upper side of the disulfide and cysteine 104 is on the lower side. Dashed lines indicate missing density from the structures. Residue positions where sequons asparagine residues are observed in AL λ LCs are shown as spheres, colored to indicate the number of sequons at each position (data from Figure 4 and Figure 8). Note that not all IMGT positions are occupied in each structure, so not all the sequon positions can be shown.

**Table 1 pharmaceuticals-17-01542-t001:** Numbers of LC sequences analyzed. Monoclonal LCs are from the AL-Base AL and MM subcategories [15].

Sequence Origin	*IGKV*			*IGLV*			Total		
	Total	Sequon	No Sequon	Total	Sequon	No Sequon	Total	Sequon	No Sequon
AL subcategory	160	67 41.9%	93 58.1%	524	39 7.4%	485 92.6%	684	106 15.5%	578 84.5%
MM subcategory	595	62 10.4%	533 89.6%	374	38 10.2%	336 89.8%	969	100 10.3%	869 89.7%
OAS repertoire	4,278,425	360,785 8.4%	3,917,640 91.6%	3,769,322	271,185 7.2%	3,498,137 92.8%	8,047,747	631,970 7.9%	7,415,777 92.1%

**Table 2 pharmaceuticals-17-01542-t002:** Cysteine sequons (NxC) in AL and MM LCs.

AL-Base Subcategory	*IGV_L_* Gene	Region	Asn Position (IMGT)	Sequence	Number of Sequences
AL	*IGLV2-23*	CDR3	114	NTC	1
AL	*IGLV3-1*	CDR1	38	NAC	2
AL	*IGLV3-1*	CDR1	38	NVC	2
MM	*IGKV1-39*	CDR1	36	NTC	1

## Data Availability

The original contributions presented in the study are included in the article, further inquiries can be directed to the corresponding author/s.

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
