# Peer review of "Predicting Structural Consequences of Antibody Light Chain N-Glycosylation in AL Amyloidosis"

_pharmaceuticals, 2024, doi:10.3390/ph17111542_

Round 1

Reviewer 1 Report

Comments and Suggestions for Authors

Reviewer report on manuscript title “Structural Consequences of Antibody Light Chain N-Glycosylation in AL Amyloidosis.”

This manuscript presents a theoretical study investigating the frequency and positioning of N-glycosylation sequence motifs (sequons) within light chain variable domain sequences. The study aims to evaluate the structural impact of N-glycosylation on light chains and its possible relationship with amyloid formation propensity in vivo. To achieve this, the authors compare amyloidogenic and myeloma monoclonal light chains from the AL-BASE database with polyclonal light chain sequences from the Observed Antibody Space.

The study finds that the increased frequency of sequons in amyloidogenic light chains is associated with specific variable domain gene segments linked to AL amyloidosis. Furthermore, the authors demonstrate that most sequons result from somatic mutations and frequently occur at hot spots, with structural positioning that differs between kappa and lambda light chains. However, no substantial difference between amyloidogenic and myeloma light chains within the same isotype was observed. Of note, in lambda light chains, sequons are often located in the CDR3 region and in residues near the interface between heavy and light chains or the interface of the light chain homodimer. This observation led the authors to hypothesize that N-glycosylation may promote amyloid aggregation of lambda light chains by destabilizing their dimeric complexes in circulation, favoring dissociation into monomers prone to misfolding and aggregation. In contrast, no clear structural rationale was identified for the association between N-glycosylation in kappa light chains and amyloidosis.

This manuscript is highly relevant to the special issue "Amyloid Composition and Structure-Based Development of Therapeutic and Diagnostic Agents" due to the potential influence of N-glycosylation on the physicochemical, biological, and structural characteristics of the circulating monoclonal light chain (AL precursor) and AL fibrils. As noted by the authors in their discussion, N-glycosylation may alter the immunological properties of AL fibrils, potentially impacting their recognition by therapeutic or diagnostic anti-AL fibril antibodies. This post-translational modification may also influence interactions with molecules that modulate fibril stability and susceptibility to proteolytic degradation.

The theoretical analysis is well-designed, employs appropriate statistical methods, and utilizes high-quality sources of light chain sequences. The manuscript is logically structured, with a well-organized presentation of background information that supports the study’s rationale. Although the introduction is somewhat lengthy, it is well-structured and covers the essential aspects of the topic. The manuscript is well-written, with accurate use of English and technical terminology. Figures and tables are thoughtfully designed and effectively illustrate the findings.

Comment:

There appears to be a potential typo in Table 1. In the "Total" section, under the "sequon" column, the value is listed as 0.79%; however, it appears that the correct value should be 7.9%.

Reviewer 2 Report

Comments and Suggestions for Authors

The manuscript presents an original and comprehensive analysis of the role of N-glycosylation in light chains (LCs) and its association with amyloid diseases such as AL amyloidosis and multiple myeloma (MM). While N-glycosylation has been discussed in the context of protein stability and folding, its specific implications for LC misfolding and amyloid formation provide novel insights. The study is scientifically sound and based on robust data analysis, including the use of the AL-Base and OAS databases, bioinformatics tools (such as ANARCI and IMGT numbering), and structural modeling tools (like ChimeraX). However, one major limitation is the reliance on in silico analyses without direct experimental validation. While the authors acknowledge this, the absence of experimental evidence (e.g., mass spectrometry-based identification of glycosylated LCs in patients) weakens the study’s conclusions. Future work incorporating experimental verification of the predicted glycosylation sites would enhance the scientific validity of the results. If the study’s hypotheses about glycosylation-related aggregation and amyloid formation are experimentally confirmed, the findings could have substantial implications for the understanding and treatment of amyloid diseases.

Major comments:

Figure 3 in the PDF version is cut off and unreadable in its current form. Please correct this.

Figure 8. Lines 347-348: “Residues are colored according to solvent exposure: grey, surface-exposed; red, buried in the core;” How did the authors determine this? Please show on the model how exactly the presented amino acid residues colored in red are buried in the core.

In the text, especially in sections 2. Results and 4. Materials and Methods, there is an extensive use of abbreviations, which may hinder the readability of the manuscript. A table of abbreviations could be helpful.

Round 2

Reviewer 2 Report

Comments and Suggestions for Authors

I believe the authors have addressed the main comments and made appropriate revisions to the manuscript. I think the manuscript could be published after minor corrections.

Minor comments:

Lines 548-549, 588-589. Please verify the relevance of reference #86. I believe reference #84 might be intended here.

Lines 581-582. Please check the relevance of reference #65, and also double-check the accuracy of references throughout the document.

Line 591. “8 × 0X” likely should be “8X0X”.

Author Response

Lines 548-549, 588-589. Please verify the relevance of reference #86. I believe reference #84 might be intended here.

Lines 581-582. Please check the relevance of reference #65, and also double-check the accuracy of references throughout the document.

Thank you for spotting these, the reference manager software had not updated the bibliography after the last round of edits. The references are now correctly numbered.

Line 591. “8 × 0X” likely should be “8X0X”.

Yes; corrected. Thank you.